# The Effect of Nighttime Snacking on Cognitive Function in Older Adults: Evidence from Observational and Experimental Studies

**DOI:** 10.3390/nu14224900

**Published:** 2022-11-19

**Authors:** Cheng-Cheng Niu, Wei-Jie Bao, Hai-Xin Jiang, Jing Yu

**Affiliations:** 1Faculty of Psychology, Southwest University, Chongqing 400715, China; 2School of Humanities and Social Science, The Chinese University of Hong Kong, Shenzhen 518172, China

**Keywords:** older adults, nighttime snacking, cognitive function, dietary habit, acute ingestion

## Abstract

Evidence shows that supplementary snacking could provide older adults with nutrients that cannot be obtained through three meals a day. However, whether and how supplementary snacking, especially nighttime snacking, affects older adults’ cognitive function remain unclear. The present study examined the effect of nighttime snacking on cognitive function for older adults. In study 1, we investigated the association between nighttime snacking and cognitive function based on data from 2618 community-dwelling older adults from the China health and nutrition survey (CHNS). In study 2, we conducted an experiment (*n* = 50) to explore how nighttime acute energy intake influences older adults’ performance on cognitive tasks (immediate recall, short-term delayed recall, and long-term delayed recall). Both the observational and experimental studies suggested that nighttime snacking facilitated older adults’ cognitive abilities, such as memory and mathematical ability, as indicated by subjective measures (study 1) and objective measures (studies 1 and 2). Moreover, this beneficial effect was moderated by cognitive load. These findings bridge the gap in the literature on the relationships between older adults’ nighttime snacking and cognitive function, providing insight into how to improve older adults’ dietary behaviors and cognitive function.

## 1. Introduction

Observational studies have shown that a healthy diet is associated with better cognitive outcomes (e.g., [1,2,3]). It was shown that older adults are at high risk of inadequate energy intake and that inadequate energy intake increases the likelihood of cognitive decline [4,5,6]. Supplementary snacking could provide older adults with nutrients that cannot be obtained through three meals per day, which might slow cognitive decline [7,8]. However, our knowledge concerning supplementary snacking, especially nighttime snacking, in older adults is still limited. Since there is a long interval between dinner and breakfast the next day, a little nighttime snacking is a common way of obtaining supplementary energy. It is thus imperative to understand the relationship between nighttime snacking and the cognitive function of older adults and to develop possible strategies from the perspective of dietary behaviors to slow cognitive decline in later life.

Previous studies have mainly focused on the effect of nighttime snacking on the physiological functions of older adults. Results showed that eating a small amount of nutrient-dense but low-energy food before sleep could promote older adults’ physical health [7,9,10]. More specifically, studies have suggested that appropriate nighttime snacking compensates for anabolic resistance and benefits protein synthesis and cardiometabolic health [10,11,12,13]. However, the role of a nighttime supply of energy in older adults’ cognitive function is less understood. To the best of our knowledge, there is little research exploring the link between nighttime snacking and the cognitive function of older adults with either subjective (e.g., validated questionnaires) or objective measures (e.g., accuracy and reaction times in cognitive tasks).

In the literature, some studies have tested the relationship between ingesting glucose or sugar and cognitive function, and the results are controversial. Some studies have suggested that chronic glucose consumption and habitual sugar intake contribute to cognitive impairment (e.g., [14,15]). Nevertheless, accumulated evidence has also shown that the acute ingestion of a certain amount of glucose increases the synthesis and release of acetylcholine and promotes the activity of the hippocampus, thereby improving individuals’ cognitive function (e.g., [16,17]). It has been well-documented that acute energy ingestion could facilitate a wide range of cognitive abilities, such as attention, visual processing, verbal memory, and episodic memory [18,19,20,21,22]. It is also worth noting that the effect of acute energy ingestion is moderated by cognitive load [18]. It was demonstrated in previous studies that acute energy consumption is more effective for high versus low cognitive resource-demanding tasks [23,24]. However, few studies have considered whether and how acute energy ingestion at night impact cognitive activities in later life.

This study aimed to examine the relationship between nighttime snacking and the cognitive function of older adults through observational and experimental studies. In study 1, we investigated the association between nighttime snacking and cognitive function manifested with objective and subjective measures based on the data from the China health and nutrition survey (CHNS), a representative national survey of community-dwelling elderly people. In study 2, we conducted an experiment to test the effect of nighttime acute energy intake on older adults’ performance on cognitive tasks and the moderating effect of cognitive load. We hypothesized the following:

**Hypothesis** **1 (H1).***Older adults with nighttime snacking habits are associated with better cognitive function*.

**H1a.** *The dietary habit of nighttime snacking is positively correlated with cognitive function indicated by objective measures*.

**H1b.** 
*The dietary habit of nighttime snacking positively predicts older adults’ cognitive function, indicated by subjective measures.*


**Hypothesis** **2 (H2).***In a quantitative and controlled experiment, nighttime acute energy intake increases older adults’ cognitive performance*. 

**H2a.** *The beneficial effect of nighttime energy intake is moderated by cognitive load and is more robust in the high-load condition*.

**H2b.** *The effect of nighttime energy intake decreases after nocturnal sleep*.

## 2. Study 1: Observational Study—Evidence from Dietary Habits

Study 1 explored the cross-sectional relationship between nighttime snacking and the cognitive function of older adults. To specify, we explored in a cross-sectional study whether and how dietary habits were associated with cognitive function gauged with both objective measures (H1a) and subjective measures (H1b).

### 2.1. Materials and Methods

#### 2.1.1. Data and Samples

The CHNS is an ongoing, open-access, prospective, community-based cohort study jointly conducted by the National Institute of Nutrition and Food Safety in China and the population center of the University of North Carolina in the USA. The multi-stage random cluster process was used to extract samples from nine provinces (i.e., Liaoning, Heilongjiang, Shandong, Jiangsu, Henan, Hubei, Hunan, Guangxi, and Guizhou) with nine waves of data. This study selected the cross-sectional data of CHNS in 2006 for the following analysis, as this wave assessed both variables that we were interested in, i.e., dietary behavior and cognitive function. The flow chart for the data selection is shown in Figure 1. Participants aged 55 years and older were included, and those lacking the necessary information (i.e., data on cognitive function, sex, and education) were excluded. Additionally, those with hypertension, diabetes, and stroke were excluded. Therefore, a total of 2618 participants were included, of which 500 (19.1%) were in the nighttime snacking group, and 2118 (80.9%) were in the non-nighttime snacking group (see the immediately following section for the assignment of participants to groups). The survey was approved by the institutional review committees of the University of North Carolina, USA, and the National Institute of Nutrition and Food Safety, China. All participants provided informed consent.

#### 2.1.2. Nighttime Snacking

In the 2006 wave, the participants provided dietary information for three consecutive days. The recalled time was randomly allocated from Monday to Sunday, and the data of each sampling unit were almost equal in the seven days of one week. Our primary predictor was whether older adults had a habit of nighttime snacking. The participants were asked about their “meal time” each day, and there were six options: (1) breakfast, (2) morning snack, (3) lunch, (4) afternoon snack, (5) dinner, and (6) evening snack. In the three-day diet recalled record, the participants who reported having an evening snack in one or more days would be classified into the nighttime snacking group (1). Otherwise, they would be classified into the non-nighttime snacking group (0).

#### 2.1.3. Cognitive Function: Objective and Subjective Measures

Cognitive function was assessed with both objective and subjective measures. The objective measures represented scores in four cognitive tasks: immediate recall (a total score of 10), delayed recall (a total score of 10), counting backward from 20 (a total score of 2), and 7 serial subtractions (a total score of 5). To specify, the sum of those four test scores [25], ranging from 0 to 27, was achieved to measure cognitive function, with higher scores indicating better cognitive function. To achieve the subjective measures, the participants were required to rate their memory function (“How is your memory?”). Their self-reported memory ability was assessed on a 5-point Likert scale with 1 = very good and 5 = very bad. We reversed the score of the self-reported memory ability to be in line with the objective measures, and thus, higher scores indicated better cognitive function.

#### 2.1.4. Covariates

Demographic characteristics were included as covariates, as well as smoking and drinking behaviors that could potentially be associated with older adults’ cognitive function (e.g., [26,27,28]). To specify, age, sex (male or female), education level (junior high school and below, senior high school, university, or above), smoking (yes or no), and drinking frequency (<1 time/week, 1–4 times/week, or almost every day) were involved as the covariates. 

#### 2.1.5. Statistical Analyses

The groups’ differences (nighttime snacking versus non-nighttime snacking) in demographics and cognitive function were tested with either the Mann-Whitney *U* test or the *χ*^2^ test. Moreover, hierarchical linear regressions were used to explore the association between nighttime snacking and cognitive function. Step 1 adjusted for age, sex, and education; step 2 further adjusted for smoking and drinking behaviors; and step 3 entered the variable of nighttime snacking. All the analyses were performed using SPSS 22.0 (IBM Corporation, Armonk, NY, USA).

### 2.2. Results

#### 2.2.1. Cognitive Differences between the Groups

A total of 2618 participants were included to analyze the variables of interest. The sample characteristics are shown in Table 1. The older adults that displayed nighttime snacking generally performed better than their non-nighttime snacking counterparts in both objective measures (Mann-Whitney *U* test, *p* < 0.001) and subjective measures of cognitive function (Mann-Whitney *U* test, *p* < 0.001).

#### 2.2.2. The Associations between Nighttime Snacking and Cognitive Function

The hierarchical linear regression models were used to investigate the associations between nighttime snacking and the objective and subjective measures of cognitive function. The variables of interest entered the regression model in three steps (Table 2). In terms of the objective measures of cognitive function, after controlling for age, sex, education, smoking, and drinking frequency, the fully adjusted model (step 3) showed a positive association between nighttime snacking and cognitive function (*β* = 0.05; *t* = 3.06; *p* < 0.01). This suggests that older adults with a habit of nighttime snacking were more likely to have a higher cognitive function. The results of analyzing the self-reported scores of cognitive function were similar: The entire model showed a positive association between nighttime snacking and self-reported memory scores (*β* = 0.07; *t* = 3.53; *p* < 0.001), indicating that older adults with a habit of nighttime snacking were higher in cognitive function. 

## 3. Study 2: Experimental Study—Evidence from Acute Ingestion

Study 2 aimed to examine, in a controlled experiment, the positive association that we observed in study 1 between nighttime snacking and the cognitive function of older adults (H2). Additionally, we manipulated the cognitive load and delay in-between the acute ingestion and cognitive tests to examine the modifying function of cognitive load (H2a) and timeliness (H2b). To that end, we conducted a 2 (drink: glucose versus placebo) × 2 (cognitive load: high- versus low-load) × 3 (delay: 0, 5-min, and 12-h) experiment. 

### 3.1. Materials and Methods

#### 3.1.1. Participants

A total of 51 older adults (age: *M* = 67.98, *SD* = 5.45, range 56–76 years, 68.6% female) from local communities were recruited. According to previous studies (e.g., [23,29,30]), 20 to 25 participants per group is enough to detect a potential effect of glucose on cognitive function. The inclusive criteria were: (1) 55 years or older; (2) had normal or corrected-to-normal hearing, as participants were required to complete an auditory memory test; (3) had no diabetes, hypertension, or hyperglycemia; and (4) scored 27 or over in the mini-mental state examination (MMSE [31]). The older adults were randomly assigned to either the glucose or the placebo group. One was excluded for failing to understand the task, which resulted in a sample of 25 for each group. The glucose and placebo groups were matched in age, sex, education, and MMSE scores (*ps* > 0.05, Table 3). 

The study was approved by the ethics committee of the Faculty of Psychology, Southwest University (H22065). The participants provided informed consent and were compensated for their participation.

#### 3.1.2. Experimental Design

A 2 (drink: glucose versus placebo) × 2 (cognitive load: high- versus low-load) × 3 (delay: 0, 5-min, and 12-h) design was applied, with the drink as a between-subject factor and cognitive load and delay as within-subject factors. The older adults in the glucose group were required to ingest a drink containing 25 g of glucose, which provided 100 kcal energy, which is approximately equivalent to a 200 g apple [23,32,33]. Meanwhile, the older adults in the placebo group were required to ingest a drink with an artificial sweetener (115 mg of aspartame).

After the drink, participants were asked to rate their pleasure on a 10-point Likert scale. The results showed that there was no significant difference in the pleasure scores between the glucose (*M* = 8.68, *SD* = 2.08) and placebo conditions (*M* = 7.52, *SD* = 2.38; *t* (48) = 1.84, *p* = 0.07, Cohen’s *d* = 0.53), indicating that the palatability was similar in the two conditions and thus had limited influence on the results of the current study. As for the cognitive load, the older adults were asked to fulfill an auditory verbal learning task (AVLT [34,35]) in the low-load condition and were simultaneously administered the AVLT and a 1-back task in the high-load condition.

#### 3.1.3. Experimental Procedure 

The experimental procedure is shown in Figure 2. The older adults were asked to arrive at the laboratory between 8:00 and 8:30 p.m. after dinner and refrain from any food and drink for two hours before arriving. The older adults were first required to complete demographic questionnaires and neuropsychological assessments and then were randomly assigned to consume a drink of either glucose or the placebo. Afterward, they were asked to provide a pleasure rating on their status. After a 10-min waiting period, the older adults were administered the low-load and the high-load AVLT tasks with a 10-min interval in between, and the sequence of tasks was counterbalanced among the participants. For instance, a participant performed on the low-load AVLT at first, and the high-load AVLT was started only if the low-load AVLT was consecutively repeated twice. After each trial (low-load or high-load) of the AVLT; namely, after the presence of a list of 15 words (see the immediately following section), the participants performed an immediate recall; and after the final trial of the low- and high-load AVLT, the participants performed a short-term delayed recall 5 min later than the immediate recall. After nocturnal sleep, the participants revisited the laboratory and performed the long-term delayed recall with no additional learning.

#### 3.1.4. Cognitive Tasks

The older adults were administered variants of the AVLT tasks. After hearing a list of 15 words, the older adults were asked to recall as many words as possible. This procedure was repeated three times. In the high-load condition, the older adults were asked to perform a 1-back task simultaneously. Specifically, a sequence of single-digit numbers was presented on the center of the screen together with auditory words, with each number being displayed for 1 s, followed by an inter-stimulus interval of 1 s. The older adults were reminded to pay equal attention to both tasks and not to prioritize one over the other. Two different word lists were presented in the low- and high-load conditions, and the allocation of the lists to the conditions was randomized. The memory performance was measured with an immediate free recall test at the end of the last study trial, a short-term delayed recall test 5 min later, and a long-term delayed recall test the next morning after nocturnal sleep.

#### 3.1.5. Statistical Analysis

A series of 2 (drink: glucose versus placebo) × 2 (cognitive load: low- versus high-load) mixed-effects analysis of variance (ANOVA), with the drink as a between-subjects factor and cognitive load as a within-subjects factor, was conducted to separately test the effect of energy intake on immediate recall (after the final trial of the high- and low-load AVLT), short-term delayed recall, and long-term delayed recall. A *Bonferroni* correction was employed for multiple comparisons. 

### 3.2. Results

#### 3.2.1. Immediate Recall

For the performance in the immediate recall test, the main effect of the cognitive load was significant (*F* (1, 48) = 117.52, *p* < 0.001, *η_p_*^2^ = 0.71), with more words being recalled in the low-load than the high-load condition. Moreover, the interaction effect of drink × cognitive load was significant (*F* (1, 48) = 6.63, *p* < 0.05, *η_p_*^2^ = 0.12; Figure 3a). Further simple-effect analyses revealed that there was no significant difference between the glucose and the placebo groups in the low-load condition (glucose: *M* = 9.48, *SD* =2.08; placebo: *M* = 9.28, *SD* = 2.41; *t* (48) = 0.31, *p* = 0.76, Cohen’s *d* = 0.09), whereas the glucose drink benefited older adults’ memory performance in the high-load condition (glucose: *M* = 7.04, *SD* = 1.86; placebo: *M* = 5.32, *SD* = 1.70; *t* (48) = 3.41, *p* < 0.001, Cohen’s *d* = 0.99). The results showed that the effect of glucose on older adults’ performance in immediate recall was moderated by cognitive load, and a facilitating effect was observed in the high-load condition only.

#### 3.2.2. Short-Term Delayed Recall

For the performance in the short-term delayed recall test, the main effect of the cognitive load was significant (*F* (1, 48) = 143.42, *p* < 0.001, *η_p_*^2^ = 0.75), with more words being recalled in the low-load than the high-load condition. Similarly, the interaction effect of drink × cognitive load was significant (*F* (1, 48) = 5.12, *p* < 0. 05, *η_p_*^2^ = 0.10; Figure 3b). Furthermore, the simple-effect analysis uncovered that there was no significant difference between the glucose and placebo groups in the low-load condition (glucose: *M* = 8.04, *SD* = 2.59; placebo: *M* = 7.89, *SD* = 2.64; *t* (48) = 0.22, *p* = 0.83, Cohen’s *d* = 0.06), whereas the glucose group recalled more words than the placebo group in the high-load condition (glucose: *M* = 5.12, *SD* = 2.15; placebo: *M* = 3.60, *SD* = 1.87; *t* (48) = 2.67, *p* < 0.01, Cohen’s *d* = 0.77). This suggests that the impact of glucose intake also benefited older adults’ memory, which remained after a short-term delay.

#### 3.2.3. Long-Term Delayed Recall

As for the performance in the long-term delayed recall test, the main effect of the cognitive load remained significant (*F* (1, 48) = 50.41, *p* < 0.001, *η_p_*^2^ = 0.51). However, the interaction effect of drink × cognitive load (*F* (1, 48) = 2.02, *p* = 0.16, *η_p_*^2^ = 0.04; Figure 3c) was no longer significant, which suggests the effect of glucose faded after nocturnal sleep. Nevertheless, further simple effect analyses showed that there was no significant difference between the glucose and the placebo groups in the low-load condition (glucose: *M* = 6.72, *SD* = 3.20; placebo: *M* = 6.52, *SD* = 3.40; *t* (48) = 0.22, *p* = 0.83, Cohen’s *d* = 0.06). However, the beneficial effect of glucose was still significant in the high-load condition after sleep (glucose: *M* = 4.16, *SD* = 2.34; placebo: *M* = 2.68, *SD* = 1.93; *t* (48) = 2.44, *p* < 0.05, Cohen’s *d* = 0.70).

## 4. Discussion

In the present study, we found that nighttime snacking facilitated older adults’ cognitive function. Study 1 indicated a positive correlation between older adults’ nighttime snacking and cognitive function, and this positive correlation was established on both objective (H1a) and subjective measures (H1b) of cognitive function. In study 2, we manipulated the nighttime acute energy intake and tested its effect on participants’ performance on subsequent cognitive tasks. The results of study 2 were in line with the finding in study 1. Moreover, it was also demonstrated that this facilitating effect was moderated by cognitive load and was present in the high cognitive load condition only (H2a), and this beneficial effect faded as time went by (H2b). These findings bridge the gap in the literature on the relationship between nighttime snacking and cognitive function in later life, providing insight into how to improve older adults’ dietary behaviors and cognitive function.

It was shown that nighttime snacking leads to obesity, poor sleep, and metabolic syndrome [36,37,38]. However, several studies on the effects of energy intake between meals on cognitive performance suggest that supplementary energy intake could effectively improve people’s selective attention and positively impact cognitive behaviors, such as learning and memory [39,40]. The amount of energy intake is a crucial factor. Previous studies have demonstrated that a small amount (~150 kcal) of a single nutrient or mixed diet before sleep is not harmful but advantageous to physical and mental health [9]. In addition, a 4-week intervention study on sedentary obese female showed that ingesting ~150 kcal of whey, casein, or carbohydrates before sleep (at least 2 h after dinner) combined with physical exercise not only reduced body fat, optimized body structure, and strength but also improved the emotional status in intervention groups [41]. Thus, the health outcomes of nighttime snacking likely depend on the kinds of population and the amount of food intake. Supplementary energy intake is typical among older adults [7]. Extant studies have shown that nighttime snacking could positively impact older adults’ physiological and emotional functions, and our study further explored the effect on the cognitive function of older adults. 

In study 2, we also found a beneficial effect of nighttime energy intake on older adults’ cognitive performance. Moreover, our findings suggest that this effect is specific to tasks requiring more cognitive resources that probably consume more energy. It is consistent with previous findings that propose that glucose particularly enhances tasks that are complex and require divided attention [18,23,24]. For instance, a beneficial effect of glucose on memory was not detected when participants encoded a 20-word list alone but was present when the participants simultaneously performed a secondary task [24]. 

In addition, the current study attempted to uncover how long the effect could survive. Our results showed that the facilitating effect of nighttime acute energy intake on cognitive function subsided gradually. To specify, the acute energy intake at night significantly improved memory performance in immediate recall and short-term delayed recall, whereas its influence was weakened in long-term delayed recall after nocturnal sleep. In contrast, a previous study investigating the effects of glucose on the memory performance of healthy young participants showed that glucose consumption had little effect on immediate free recall but facilitated their performance on delayed recall [42]. One possible explanation for that discrepancy is that the delays in short-term and long-term conditions varied across studies. In the present study, the beneficial effect decayed after nocturnal sleep (approx. 12 h). However, in Foster et al.’s study, the long-term condition was administered only 20 min after the short-term condition. Moreover, Foster used different memory tasks: a free recall task in immediate recall, a cued-recall task in short-term delayed recall, and a recognition task in long-term delayed recall. In the present study, we used free recall for all three tasks. Different memory tests may blur how the effect of glucose varied with time.

Several limitations should still be considered. First, total energy and diet quality are important factors in diet-related studies. Some studies showed that total energy and diet quality impact the cognitive function of older adults [43,44]. However, the open-access database used in study 1 did not provide relevant information, so we could not examine the role of total energy and diet quality. Further research is needed to learn more about how to administer nighttime snacking (e.g., how to select which types of food to ingest and how much total energy is needed) to maximize the beneficial effect for older adults. Second, the present study involved healthy older adults only. However, individual differences in physiological and cognitive functions are large between elderly people of the same age (70+) [45,46]. Since health problems alongside aging are common and prominent, such as obesity, diabetes mellitus, high blood pressure, heart disease, and food allergies, we should be cautious when generalizing the present findings to a much wider range of the population. Third, just a small set of cognitive measures and abilities have been considered in the present study; how the effect of nighttime snacking applies to other cognitive domains, such as processing speed, attention, and executive function, needs to be further explored.

## 5. Conclusions

The current studies uncover the relationship between nighttime snacking and cognitive function in later life. Our work suggests that nighttime snacking could promote older adults’ cognitive function. It casts light on a potential avenue for delaying older adults’ cognitive decline and preventing the development of cognitive disorders. However, we are still far from recommending nighttime snacking, especially when individual and food differences are considered. Informative research and further evidence are needed to reveal the underlying mechanisms of how nighttime snacking affects older adults’ cognitive function. 

## Figures and Tables

**Figure 1 nutrients-14-04900-f001:**
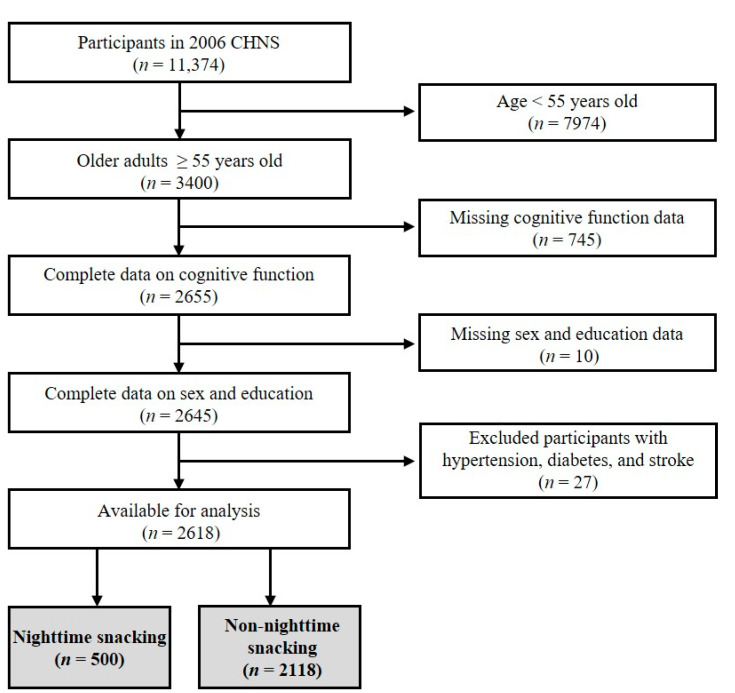
The flow chart of participant selection.

**Figure 2 nutrients-14-04900-f002:**
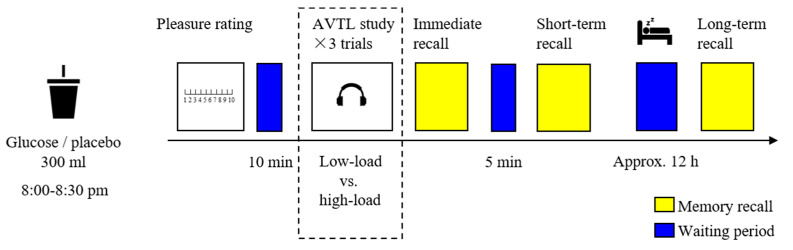
The experimental procedure. After consuming a drink of glucose or placebo, the older adults were administered both low- and high-load AVTL tasks. The immediate recall, short-term delayed recall, and long-term delayed recall were tested at different time points.

**Figure 3 nutrients-14-04900-f003:**
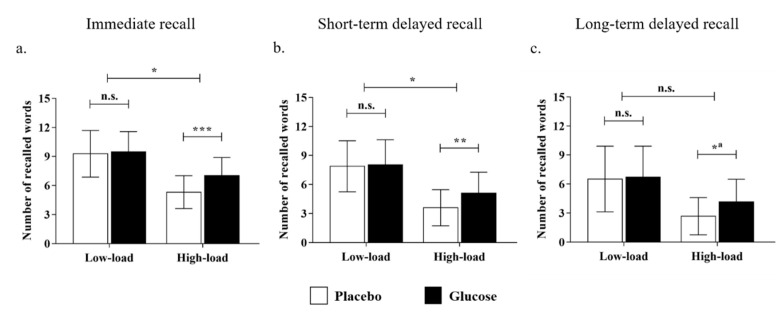
The number of recalled words out of 15 after the intake of a drink (glucose/placebo) in the low- and high-load conditions for the (**a**) immediate recall, (**b**) short-term delayed recall, and (**c**) long-term delayed recall. *Note: * p* < 0.05; *** p* < 0.01; **** p* < 0.001; n.s. = non-significant. ^a^ The interaction effect was not significant. Nevertheless, a simple effect analysis was conducted to show the tendency.

**Table 1 nutrients-14-04900-t001:** Nighttime snacking versus non-nighttime snacking.

	Nighttime Snacking (*n* = 500)	Non-Nighttime Snacking (*n* = 2118)	*p*-Value
Age (years)	65.00 (7.64)	64.70 (7.83)	0.28 ^b^
Sex (female)	257 (51.4%)	1039 (49.1%)	0.37 ^a^
Education			<0.001 ^a^
Junior high school and below	336 (67.2%)	1836 (86.7%)	
Senior high school	62 (12.4%)	109 (5.1%)	
University and above	102 (20.4%)	173 (8.2%)	
Smoking			0.41 ^a^
Yes	169 (33.8%)	760 (35.9%)	
No	331 (66.2%)	1358 (64.1%)	
Drinking			0.30 ^a^
<1 time/week	387 (77.7%)	1600 (76.0%)	
1–4 times/week	39 (7.8%)	213 (10.1%)	
Almost every day	72 (14.5%)	292 (13.9%)	
Sum score in 4 cognitive tasks (0~27)	15.38 (5.00)	14.04 (6.00)	<0.001 ^b^
Self-reported memory score (1~5)	3.17 (0.98)	2.96 (0.89)	<0.001 ^b^

*Note:*^a^ *χ*^2^ test was conducted; ^b^ Mann-Whitney *U* test was conducted.

**Table 2 nutrients-14-04900-t002:** Hierarchical regression analysis of nighttime snacking on cognitive function (*n* = 2618).

	Objective Measure of Cognitive Function				Subjective Measure of Cognitive Function			
	*Β (SE)*	*t*	Δ*R*^2^	Δ*F*	*Β* (*SE*)	*t*	Δ*R*^2^	Δ*F*
*Step 1*								
Age	−0.36(0.01)	−20.58 ***	0.21	231.82 ***	−0.22(0.002)	−11.83 ***	0.09	81.11 ***
Sex	0.12(0.21)	6.61 ***			0.07(0.04)	3.69 ***		
Education	0.23(0.16)	13.10 ***			0.16(0.03)	8.35 ***		
*Step 2*								
Smoking	0.02(0.27)	1.03	0.001	0.85	−0.003(0.05)	−0.15	0.001	1.61
Drinking	0.01(0.16)	0.56			0.04(0.03)	1.78		
*Step 3*								
Nighttime snacking	0.05(0.26)	3.06 **	0.003	9.37 **	0.07(0.04)	3.53 ***	0.004	12.47 ***

*Note: ** p* < 0.01; **** p* < 0.001.

**Table 3 nutrients-14-04900-t003:** Characteristics of the sample in two drink groups.

	Placebo *n* = 25	Glucose *n* = 25	*p*-Value
Age (years)	69.2 (4.85)	66.84 (5.93)	0.13
Sex (M/F) ^a^	7/18	9/16	0.54
Education (years)	9.88 (2.79)	9.04 (2.09)	0.23
MMSE	28.16 (1.21)	28.52 (1.26)	0.31

*Note:*^a^ *χ*^2^ test was conducted. MMSE = the mini-mental state examination.

## Data Availability

The data that support the findings are available in the open science framework https://osf.io/r2bv3/ accessed on 27 August 2022.

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
