# Peer review of "The Effect of Nighttime Snacking on Cognitive Function in Older Adults: Evidence from Observational and Experimental Studies"

_nutrients, 2022, doi:10.3390/nu14224900_

Round 1

Reviewer 1 Report

I really like the presented manuscript. I have a few remarks, which are listed below:

1. The authors focus on the acute effects of glucose intake and stress its positive impact on cognition. There is growing literature concerning chronic glucose consumption and its adverse effects on the hippocampus and cognitive functions (e.g. animal study: Noble, E. E., Hsu, T. M., Liang, J., & Kanoski, S. E. (2019). Early-life sugar consumption has long-term negative effects on memory function in male rats. Nutritional neuroscience22(4), 273-283. human study: Chong, C. P., Shahar, S., Haron, H., & Din, N. C. (2019). Habitual sugar intake and cognitive impairment among multi-ethnic Malaysian older adults. Clinical interventions in aging14, 1331. I encourage authors to add this alternative view on sugar consumption to their introduction.

2. I would suggest using nonparametric statistical tests in study 1 as groups being compared differ in N (500 vs over 2000, t-test is not appropriate in such case).

3. The manuscript requires proofreading, some typos and grammatical mistakes are present.

Author Response

I really like the presented manuscript. I have a few remarks, which are listed below:

Point 1: The authors focus on the acute effects of glucose intake and stress its positive impact on cognition. There is growing literature concerning chronic glucose consumption and its adverse effects on the hippocampus and cognitive functions (e.g., animal study: Noble, E. E., Hsu, T. M., Liang, J., & Kanoski, S. E. (2019). Early-life sugar consumption has long-term negative effects on memory function in male rats. Nutritional neuroscience, 22(4), 273-283. human study: Chong, C. P., Shahar, S., Haron, H., & Din, N. C. (2019). Habitual sugar intake and cognitive impairment among multi-ethnic Malaysian older adults. Clinical interventions in aging, 14, 1331. I encourage authors to add this alternative view on sugar consumption to their introduction.

Response 1: Thank you for the comment and suggestion. As suggested, we added this alternative view in the introduction to illustrate the pros and cons for the effect of glucose intake on congitive fucntion (p. 2, line 55-56).

Point 2: I would suggest using nonparametric statistical tests in study 1 as groups being compared differ in N (500 vs over 2000, t-test is not appropriate in such case).

Response 2: Thank you for the suggestion. In the revision, we use Mann-Whitney U test instead of t-test in Study 1 to compare group differences (p. 4-5).

Point 3: The manuscript requires proofreading, some typos and grammatical mistakes are present.

Response 3: Thank you for the suggestion. The manuscript has been edited for language by a professional in this round of revision.

Reviewer 2 Report

Thanks to the editor for the invitation. In this study, the authors have investigated the association between nighttime eating and cognitive function in older adults. This is an interesting topic, whereas there are several major limitations in this study. Please see my comments below.

1, The cross-sectional study design makes the results difficult to interpret.

2, How did you define the “small eating”? The questionnaire only provides information on the “evening snack”.

3, Have you considered the effects of total energy and quality of diet on results?

4, Low Education level is important risk factor for cognitive decline. A stratified analysis by education level is necessary.

Round 2

Reviewer 2 Report

I will keep my previous recommedation.

Author Response

We thoroughly and carefully edit our language. As reviewer #2 has not raised up detailed comments or suggestions in this round of review, we have no point-to-point replies.